# Genomic Regions and Candidate Genes Linked to Capped Hock in Pig

**DOI:** 10.3390/life11060510

**Published:** 2021-05-31

**Authors:** Lyubov Getmantseva, Maria Kolosova, Faridun Bakoev, Anna Zimina, Siroj Bakoev

**Affiliations:** 1Federal Research Center for Animal Husbandry Named after Academy Member L.K. Ernst, 142132 Dubrovitsy, Russia; m.leonovaa@mail.ru (M.K.); bakoevfaridun@yandex.ru (F.B.); siroj1@yandex.ru (S.B.); 2Department of Biotechnology, Don State Agrarian University, 346493 Persianovski, Russia; 3Centre for Strategic Planning and Management of Biomedical Health Risks, 123182 Moscow, Russia

**Keywords:** pig, capped hock, selection signatures, genome, candidate genes, A2ML1, ROBO2, MSI1

## Abstract

Capped hock affects the exterior of pedigree pigs, making them unsalable and resulting in a negative impact on the efficiency of pig-breeding centers. The purpose of this paper was to carry out pilot studies aimed at finding genomic regions and genes linked to the capped hock in pigs. The studies were carried out on Landrace pigs (n = 75) and Duroc pigs (n = 70). To identify genomic regions linked to capped hock in pigs, we used smoothing FST statistics. Genotyping was performed with GeneSeek^®^ GGP Porcine HD Genomic Profiler v1 (Illumina Inc, San Diego, CA, USA). The research results showed 70 SNPs linked to capped hock in Landrace (38 SNPs) and Duroc (32 SNPs). The identified regions overlapped with QTLs related with health traits (blood parameters) and meat and carcass traits (fatness). In total, 31 genes were identified (i.e., 17 genes in Landrace, 14 genes in Durocs). Three genes appeared in both the Landrace and Duroc groups, including A2ML1 (SSC5), ROBO2 (SSC13), and MSI1 (SSC14). We identified genomic regions directly or indirectly linked to capped hock, which thus might contribute to identifying genetic variants and using them as genetic markers in pig breeding.

## 1. Introduction

Limb diseases and specifically limb weakness (osteochondrosis) in pigs can lead to large economic losses due to a decrease in productivity [1,2,3,4,5]. Moreover, one of the serious problems facing pig farmers is the spread of bursitis of the hock and capped hock [6,7,8]. Bursitis of the hock usually occurs beneath the hock of the hind limb; it is less common in the forelimb. The capped hock is considered to be similar to the hock joint lesion, and therefore studies of capped hock and bursitis of the hock are commonly combined [6]. Capped hock is usually only a defect and does not lead to lameness, but this greatly affects the exterior of breeding pigs. This is because capped hock makes them unsalable, which has an extremely negative effect on the efficiency of pig breeding centers. We can consider production technology and genetics as predisposing factors of capped hock etiology. Revealing genome regions directly or indirectly linked to the capped hock problem may help identify genetic variants and use them as genetic markers in the selection of pig breeding stock. One approach for finding candidate genes that can become selectively valuable traits in farm animals is to identify selection signatures [9]. The discovery of selection signatures and the identification of candidate genes can identify the main genes responsible for the selected traits. The advantage of this approach is that it does not depend on the phenotype information for individual animals. Moreover, it is applicable to relatively small populations [10]. Therefore, we decided to apply the selection signatures approach to find genomic regions and genes linked to pigs’ capped hock.

## 2. Materials and Methods

In this study, we did not implement anesthesia or euthanasia, nor was any animal sacrificed. Further, this study did not involve any endangered or protected species. According to the standard monitoring procedures and guidelines, the participating holding specialists collected tissue samples following the ethical protocols outlined in the Directive 2010/63/EU (2010). The samples of pig ears (ear pluck) were obtained as a general monitoring procedure, as it is a standard practice in pig breeding [11].

The studies were carried out on Landrace pigs (n = 75) and Duroc pigs (n = 70) born in 2020. We chose Landrace and Duroc pigs because although they have different performance traits, they are both subject to capped hock. All pigs on the farm were raised under the same conditions, i.e., on concrete slotted floors. The animals were evaluated when they reached a weight of 100–110 kg (via visual inspection). The pigs were divided into two groups depending on the condition of their limbs. For groups I and II, the presence/absence of bumps in the hock region on the hind limbs of the pigs were inspected. Group I included Landrace (n = 37) and Duroc (n = 30), whereas group II included Landrace (n = 38) and Duroc (n = 40). To study population structure, we performed a singular value decomposition (SVD) of the GRM in R [12,13]. Figure 1 shows the SVD analysis. Pre-defined breed groups (Landrace and Duroc) corresponded to well-separated clusters. No outliers were apparent on the SVD plot and no stratification of pigs with or without capped hock was detected.

To determine selection signatures, we used the FST smoothing approach by comparing pigs (group I vs. group II)) in accordance with breed (Landrace (L_I vs. L_II) and Duroc (D_I vs. D_II)).

### 2.1. Genotyping

Genomic DNA was extracted from ear samples using a DNA-Extran-2 reagent kit (OOO NPF Sintol, Russia) following the manufacturer’s protocol. The quantity, quality, and integrity of DNA were assessed using a Qubit 2.0 fluorometer (Invitrogen/Life Technologies, Carlsbad, CA, USA) and a NanoDrop8000 spectrophotometer (ThermoFisher Scientific, Waltham, MA, USA). The samples were genotyped using the GeneSeek^®^ GGP Porcine HD Genomic Profiler v1 (Illumina Inc., San Diego, CA, USA), which includes 68,516 SNPs evenly distributed genes with an average spacing of 25 kb. Genotype quality control and data filtering were performed using PLINK 1.9. After excluding SNPs with a missing sample frequency of >2%, a Hardy–Weinberg equilibrium (HWE) *p*-value < 1 × 10^−7^, rel-cutoff 0.75, and LD pruned (50 5 0.2) 43,118 SNPs for Landrace pigs and 42,256 SNPs for Duroc pigs were retained for further analysis.

### 2.2. Data Processing

To study population structure, we performed a singular value decomposition (SVD) decomposition of the GRM using the SVD function in R [12,13] (Appendix A). To determine selection signatures we used smoothing FST statistics based on the proposed model with pure drift [14]. Individual SNP FST values were grouped into genomic windows (contiguous regions) to determine smoothed FST values that could identify genomic regions with high FST values. FST was performed by comparing pigs L_I vs. L_II and D_I vs. D_II. SNP regions with smoothed FST values above the 99th quantile were identified. Further, the gene and QTL content of each region was analyzed in the Ensembl genome browser (Sscrofa 11.1) (https://www.ensembl.org/index.html) (assessed on 28 May 2021). For every identified gene, we performed the PANTHER (http://www.pantherdb.org/) (assessed on 28 May 2021). enrichment analysis using the Fisher’s exact false detection rate adjustment (FDR) test. Moreover, we conducted a manual literature search for data on their associations with any traits in humans and animals.

## 3. Results

Using the smoothing FST method in pigs L_I vs. L_II, we found genome regions with strong outliers, which were comprised of 38 SNPs (Appendix A). These regions overlapped with QTLs1 (n = 147), which represent traits related with health traits (109 QTLs-blood parameters and immune capacity), meat and carcass traits (15 QTLs-anatomy, fatness, meat color and texture), reproductive traits (14 QTLs-litter traits and reproductive organs), exterior traits (7 QTLs-behavioral, defects, conformation, and coat characteristics) and production traits (2 QTLs-bodyweight and days to 100 kg) (Appendix A). In these areas, 17 genes were identified (Table 1).

On the basis of the enrichment analysis, we identified two main pathways. The first was the cadherin signaling pathway (*p*-value = 8.95 × 10^−2^), which is involved in many biological processes such as development, neurogenesis, cell adhesion, and inflammation. The second was the Wnt signaling pathway (*p*-value = 1.88 × 10^−1^), which is one of the intracellular signaling pathways for animals that regulates embryogenesis, cell differentiation, and the development of malignant tumors.

In total, 32 SNPs were identified in the genome region with strong outliers in D_I vs. D_II pigs (Appendix A). These areas overlapped with 49 QTLs. Similar to Landrace, we distinguished QTLs for meat and carcass trait: 32 QTLs, 21 of which were associated with fatness (intramuscular fat content); reproductive traits (7 QTLs-litter traits and reproductive organs), Health traits (5 QTLs-blood parameters and disease susceptibility), production traits (3 QTLs-growth and feed intake), and exterior traits (2 QTLs-conformation) (Appendix A). In these regions, we identified 14 genes (Table 2).

On the basis of the enrichment analysis, two pathways were identified. The first was the notch signaling pathway (NUMB) (*p*-value = 2.55 × 10^−2^), which provides local intercellular communication and coordinates the signaling cascade. Moreover, it plays a certain role in embryogenesis and somitogenesis. The second was the nicotinic acetylcholine receptor signaling pathway (*p*-value = 5.81 × 10^−2^), which participates in two main types of neurotransmission: synaptic (e.g., neurotransmitter release) and paracrine transmission.

Overall, 31 genes were identified in the genome region with strong outliers (17 genes in Landrace pigs and 14 genes in Duroc pigs). The A2ML1 (SSC5), ROBO2 (SSC13), and MSI1 (SSC14) genes manifested in both the Landrace and Duroc groups.

## 4. Discussion

Capped hock does not affect the welfare of pigs, but it does damage farming in terms of the sales of pedigree animals. The emergence and spread of capped hock is associated with changes in keeping technology, specifically with the type of floors [2]. There is little data that investigates the prevalence of capped hock in pig farms, but the available data proves that it does not exceed 20% [1,6,15].

Here, we conducted pilot studies to identify genomic loci and candidate genes that might be linked to the capped hock. The identified regions overlapped with QTLs loci, which is related with meat and carcass traits. Landrace pigs exhibited the most intramuscular fat content in this category. Intramuscular fat content is associated with higher nutritional qualities of pork, and therefore this trait is relevant for selection [16]. One of the factors that influences lipid accumulation and fatty acid composition is adipogenetic capacity [17]. In general, adipose tissue plays a key role in various metabolic processes and affects food intake, inflammatory response, and the meat and reproductive performance in pigs [18,19]. Therefore, lipid and fatty acid composition have a direct or indirect relationship with capped hock. Our results show that the identified regions also overlapped with QTLs responsible for health traits and mainly blood parameters. This may indicate that a predisposition to capped hock can be realized through genetic variants associated with hematological signs (concentration and content of corpuscular hemoglobin, platelet count), liver function (bilirubin concentration), kidneys (serum urea level), and susceptibility to the PRRS virus.

Among the identified genes, A2ML1 (SSC5), which encodes the protein of the alpha-macroglobulin superfamily, was identified in Landrace and Duroc pigs. The protein represents an N-glycosylated monomeric protein that behaves as an inhibitor of several proteases. These proteins display a unique trapping inhibition mechanism. Mutations in the A2ML1 gene cause the Noonan-like syndrome with a different phenotype ranging from severe (leading to intrauterine fetal death) to mild [20], as well as some cases of otitis media [21]

ROBO2 (SSC13) was also identified in the two groups. The ROBO2 gene encodes a transmembrane receptor and is a member of the immunoglobulin superfamily [22]. Four Robo homologs have been identified in mammals: Robo1 (Dutt1), Robo2, Robo3 (Rig1), and Robo4 (magic roundabout). ROBO receptor expression is crucial for axon control, cell migration, and SLIT/ROBO signaling [23]. The SLIT/ROBO complex is involved in regulating the central nervous system and is involved in lung, kidney, and heart development [24,25,26]. In addition, the SLIT/ROBO signals affect the respiratory, reproductive, immune, and circulatory systems [27]. Several studies have shown that ROBO2 gene variants are associated with immunity traits in chickens [27]. Whole-genome association studies of hematological and clinical/biochemical blood traits in large white pigs have identified a number of genes, one of which was ROBO2. It therein showed an association with hemoglobin [28]. Overall, based on our functional characterization and literature data, we determined that genetic variants of the ROBO2 gene may be associated with the capped hock phenotype in pigs.

The MSI1 (SSC14) and MSI2 (SSC12) genes are extremely interesting. The MSI2 gene is found both in Landrace and Duroc pigs, and the MSI1 gene is found only in Landrace pigs. These genes belong to a unique family of RNA-binding proteins (Musashi family) and are involved in RNA metabolism [29]. In addition, proteins of the Musashi family are considered to perform important functions in the nervous system, as well as hematopoietic systems and systems in the gastrointestinal tract (e.g., neural, hematopoietic, and gastrointestinal) in various species [30,31,32]. MSI1 functions as a regulator to maintain stem cell condition, differentiation, and tumorigenesis [33]. Excessive expression of MSI1 leads to cell proliferation and apoptosis [34]. In addition, MSI1 is associated with many malignant neoplasms in humans [35]. Many studies have recently sought to determine the molecular basis of MSI2 activity and its physiological manifestations. According to Bennett et al. [32], MSI2 is a regulator of keratinocyte migration and epithelial growth. It is also involved in regulating focal adhesion and apoptosis.

Dysregulation of adhesion molecules often leads to various diseases, including inflammation [36,37]. Cadherins play an important role in regulating adhesion because they are calcium-dependent glycoproteins. Our studies identified the CDH23 gene (cadherin 23) and the cadherin signaling pathway in Landrace pigs. In addition to the formation of strong intercellular contacts, cadherins initiate various intracellular signaling cascades and may be associated with tumor progression and inflammatory arthritis in humans [38].

Moreover, MSI showed transducing MSI/NUMB/Notch signals (Notch signaling pathway). Our studies also identified the NUMB gene (NUMB endocytic adapter protein, SSC7) and the Notch signaling pathway in Duroc pigs. Notch signaling is evolutionarily conservative and is critical during the development and homeostasis of mature tissues by regulating cell proliferation, differentiation, and cell apoptosis. In addition, the interaction of MSI/NUMB/Notch is considered to be associated with many malignant neoplasms in humans [39].

The research results showed 70 SNPs associated with the capped hock in Landrace (38 SNPs) and Duroc (32 SNPs), as well as 31 genes involved in various physiological processes in the body, including those related with inflammation, the formation of various neoplasms, and tumors. However, further research is needed to confirm the obtained results, especially research that examines other breeds and populations. The use of various methods, including the genome-wide association study, will contribute to the identification of genomic regions and candidate genes associated with capped hock, which will thus promote the production of highly productive pigs free from various defects.

## 5. Conclusions

Here, pilot studies aimed at searching for genomic loci and candidate genes linked to the capped hock in Landrace and Duroc pigs were carried out. The identified genes are involved in various physiological processes in the organism. However, it is interesting to note that the identified regions overlap with QTLs related with health traits (blood parameters), as well as in meat and carcass traits (Fatness). This may indicate that the predisposition to the capped hock can be realized through genetic variants related with composition of lipids and fatty acids, as well as in hematological traits and susceptibility to disease.

## Figures and Tables

**Figure 1 life-11-00510-f001:**
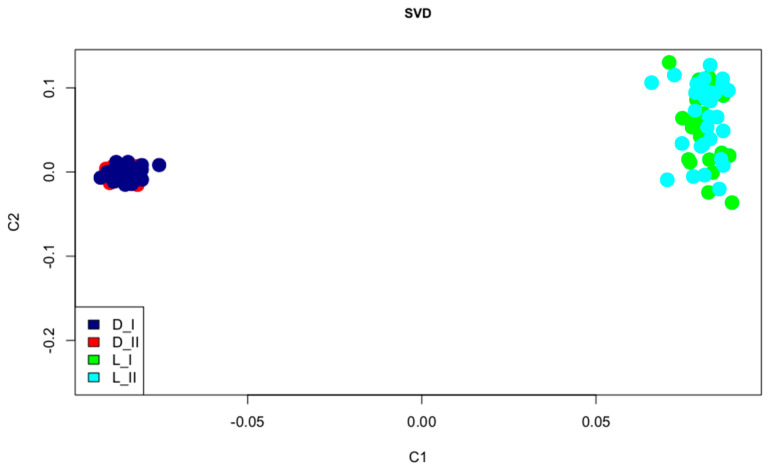
SVD plot.

**Table 1 life-11-00510-t001:** Identified genes in L_I vs. L_II.

Symbol	Full Name	SSC	Location
SASH1	SAM and SH3 domain containing 1	1	17306364..17663117
NAV2	Neuron navigator 2	2	39263939..40083539
DTWD2	DTW domain containing 2	2	122680254..122937002
SIL1	SIL1 nucleotide exchange factor	2	140884170..141118201
A2ML1	Alpha-2-macroglobulin like 1	5	62603260..62648170
HS6ST3	Heparan sulfate 6-O-sulfotransferase 3	11	65516180..66193410
MBNL2	muscle blind-like splicing regulator 2	11	66481923..66641653
MSI2	Musashi RNA binding protein 2	12	33537163..33974845
ROBO2	Roundabout guidance receptor 2	13	177365712..179040027
RIMBP2	RIMS binding protein 2	14	24394793..24722237
MSI1	Musashi RNA binding protein 1	14	40331371..40356793
SVOP	SV2 related protein	14	41829649..41931359
SART3	Spliceosome associated factor 3, U4/U6 recycling protein	14	42223752..42259992
SGSM1	Small G protein signaling modulator 1	14	42882101..42974755
CDH23	Cadherin related 23	14	74267547..74734623
PID1	Phosphotyrosine interaction domain containing 1	15	130080821..130215548
DNER	Delta/notch-like EGF repeat containing	15	130414159..130754043

**Table 2 life-11-00510-t002:** Identified genes in D_I vs. D_II.

Symbol	Full Name	SSC	Location
RBFOX1	RNA binding fox-1 homolog 1	3	34938703..37209772
A2ML1	Alpha-2-macroglobulin like 1	5	62603260..62648170
NUMB	NUMB endocytic adaptor protein	7	96624270..96798690
KCNK2	Potassium two pore domain channel subfamily K member 2	9	128429232..128659662
DTL	Denticleless E3 ubiquitin protein ligase homolog	9	131225840..131275426
FLT1	FMs-related receptor tyrosine kinase 1	11	5620698..5797095
MSI2	Musashi RNA binding protein 2	12	33537163..33974845
ZNF385D	Zinc finger protein 385D	13	8234734..9214628
SCN10A	Sodium voltage-gated channel alpha subunit 10	13	23481537..23570454
SCN11A	Sodium voltage-gated channel alpha subunit 11	13	23602378..23692281
WDR48	WD repeat domain 48	13	23735566..23786312
TTC21A	Tetratricopeptide repeat domain 21A	13	23797518..23830334
ROBO2	Roundabout guidance receptor 2	13	177365712..179040027
MYO10	Myosin X	16	5907111..6145485

(SSC–Sus scrofa chromosome).

## Data Availability

The raw data supporting the conclusions of this article will be made available by the authors upon reasonable request.

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
