# Peer review of "Genomic Regions and Candidate Genes Linked to Capped Hock in Pig"

_life, 2021, doi:10.3390/life11060510_

Round 1

Reviewer 1 Report

General remarks

The manuscript under review is a pilot study aiming to identify genomic regions associated with capped hock condition in pigs. For this purpose, the researchers categorized Landrace and Duroc animals in case/control groups based on presence or absence of bumps in the hind limbs of the animals. Next, they genotyped these animals for ~70K SNP markers and applied allele frequency-based Fst statistics, followed by annotation of the genomic regions showing extreme allele-frequency difference between case and control groups in Landrace and Duroc pigs. While the objective of the study was clear, I have following major issues with the study:

(1). First of all, genetic parameters related to capped hock condition in pigs was not discussed at all, i.e. what is the heritability estimate of the condition? Whether any previous studies have indicated any genetic basis of this condition?

(2).  What was the genetic composition of the populations under study? Did the authors assess the underlying genetic structure in both the populations? Genetic structure is major confounding factor in case-control association study (https://journals.plos.org/plosgenetics/article?id=10.1371/journal.pgen.0010032)

(3). Did author take into account all the external factors such as type of flooring and bedding that might affect the prevalence of the disease? Usually, non-linear regression models (such as logistic regression model) should be devised so as to take into account possible co-variates and other factors that might affect the prevalence of the condition.

Therefore, in my opinion, this manuscript does not satisfy the scientific rigor to get accepted in its present form.

Author Response

Reviewer 1.

Thank you for your attention to our work.

First of all, genetic parameters related to capped hock condition in pigs was not discussed at all, i.e. what is the heritability estimate of the condition? Whether any previous studies have indicated any genetic basis of this condition?

The specific causes of capped hock are still unknown, but there is little doubt that capped hock is multifactorial in origin and that genetic components play a role in its pathogenesis (PigQTLdb https://www.animalgenome.org/cgi-bin/QTLdb/SS/index).

We did not assess the genetic parameters of the capped hock, since the farms did not keep records for this trait. We did not assess the genetic parameters of the sapeda, since the farms did not keep records for this trait. For this work, we ourselves selected the animals and recorded the data.

What was the genetic composition of the populations under study? Did the authors assess the underlying genetic structure in both the populations? Genetic structure is major confounding factor in case-control association study (https://journals.plos.org/plosgenetics/article?id=10.1371/journal.pgen.0010032)

То study population structure, we performed a singular value decomposition (SVD) decomposition of the GRM using the SVD function in R (Barker et al., 2001; VanRaden, 2008). We have not used association study. We decided to apply the selection signatures approach to find genomic regions and genes linked to pigs’ capped hock. To determine selection signatures we used smoothing FST statistics based on the proposed model with pure drift (Nicholson et al. 2002). FST value of a locus is calculated as a ratio of the variance of allele frequencies between the populations and the sum of the variances within and between populations. Positive selection is indicated by high FST values ​​relative to their heterozygosity (Weigand & Leese, 2018). Smoothing of FST is used to identify contiguous genomic regions under selection. The smoothed FST method is based on the pure drift model of Nicholson et al. (2002) (Nicholson et al., 2002). According to this model, individual SNPs are grouped into genomic windows, and their average smoothed FST values ​​are calculated. Smoothed FST is useful for analyzing distantly related populations and reveals subtle differences between them (Porto-Neto et al., 2013).

 Did author take into account all the external factors such as type of flooring and bedding that might affect the prevalence of the disease? Usually, non-linear regression models (such as logistic regression model) should be devised so as to take into account possible co-variates and other factors that might affect the prevalence of the condition.

Yes, we took into account all external factors. Groups of animals were specially formed for the tasks of the study. All pigs were of the same sex, same year of birth, of the same age  and were kept in the same conditions.

Reviewer 2 Report

General comments: this paper describes a study looking at genomic regions/candidate genes for capped hock in pigs. The manuscript could be improved through amending the beginning of the introduction, as it wasn’t clear to me what the authors were trying to say. Also it would be useful for context of the study if some additional information related to the pigs used in this study were included (relatedness, diets fed up to and during the study, what type of flooring were the pigs reared on etc.). The author’s mention the impact of capped hock on the physiology of the pigs, and the genomic results indicate QTLS related to blood parameters and immune system, were any blood samples collected from the pigs and analysed for any hormones/peptides, or were any immune assays undertaken, e.g. haematology analyses, or were there any additional physiological parameters of the pigs assessed?

Also please use italics for gene names throughout and please insert p-value significance results as there didn’t seem to be any included within the manuscript.

Specific comments:

Suggest altering the title to ‘genomic regions and candidate genes linked to capped hock in pig’

Abstract:

Line 12: use past tense, ‘the purpose of this work was to carry’

Line 14: remove extra ).

Line 16: insert space at end of line

Line 17: what was the significance threshold used?

Line 19: 31 genes identified that overlapped with QTLs? It isn’t clear within the abstract why 31 genes here, but 70 genes on line 17?

Line 20: suggest changing to ‘groups including A2ML1’

Line 21: should it be ‘genomic’ instead of ‘genome’?

Introduction:

Lines 27-31: not clear if these are all meant to be one sentence, these lines should be divided into separate sentences for clarity. The first sentence here is also somewhat confusing in relation to the use of terms like ‘pigs’ organisms’? do the author’s mean pigs’ physiological status?

Line 50: remove full stop before reference

Methods:

Line 61: Why were Landrace and Duroc breeds used for this study, is capped hock more prevalent in these breeds as opposed to others?

Line 62: were all pigs used in this study of similar age, despite all being born in the same year. Was there any information about relatedness of the pigs, e.g. same boar?

Line 63: how were the pigs housed? Was there a difference in ground surface up to separation of pigs into groups that may have impacted capped hocks? Or was there a difference up to the end of the trial in the type of flooring the pigs were reared on?

Results:

Line 102: gene names should be in italics

Lines 103-108/ 116-120: what are the p-value significance of each of these pathways? And why is a pathway listed as ‘unclassified’ on line 107?

Line 118: remove ) from line

Line 121: what is meant by ‘genome regions with strong outliers’?

There are no p-values included within the results text or tables (including supplemental tables)??

Line 128: authors mention the importance of flooring type to capped hock but don’t mention flooring type used for the pigs in this study from birth within the methods

Line 168: should this be ‘capped hock’ and not ‘sarred hock’?

Line 186: in which breed?

Line 191/192: in which breed?

Line 204: should this not read ‘highly productive pigs free from various defects’ as in they don’t have the defects?

Table 1 and supplemental tables: please define SSC abbreviation

Author Response

Reviewer 2.

Thank you for your attention to our work.

The author’s mention the impact of capped hock on the physiology of the pigs, and the genomic results indicate QTLS related to blood parameters and immune system, were any blood samples collected from the pigs and analysed for any hormones/peptides, or were any immune assays undertaken, e.g. haematology analyses, or were there any additional physiological parameters of the pigs assessed?

We plan to carry out these studies in subsequent works

Specific comments:

Suggest altering the title to ‘genomic regions and candidate genes linked to capped hock in pig’

Agree with suggest. Made changes.

Abstract:

Line 12: use past tense, ‘the purpose of this work was to carry’

Made changes

Line 14: remove extra ).

Made changes

Line 16: insert space at end of line

Made changes

Line 17: what was the significance threshold used?

SNP regions with smoothed FST values above the 99th quantile were identified; the QTL content of each region was analyzed in the Ensembl genome browser (Sscrofa 11.1) (https://www.ensembl.org/index.html).

Search of QTL was performed in the Ensembl genome browser (Sscrofa 11.1) -- Ensembl Variant Effect Predictor (VEP) --  Either paste data (pfuhe;f.ncz ) -- Additional configurations -- Phenotype data and citations/

The Ensembl genome browser defines the QTL according to the Pig Quantitative Trait Locus (QTL) Database (Pig QTLdb) (https://www.animalgenome.org/cgi-bin/QTLdb/SS/index)/

Line 19: 31 genes identified that overlapped with QTLs? It isn’t clear within the abstract why 31 genes here, but 70 genes on line 17?

70 SNP were identified. However, SNPs can be localized in genes or intergenic regions. In addition, we separately checked in which regions of QTLs these SNPs are located (QTL is not always a specific gene). Accordingly, the number of SNPs, genes and QTL is different.

Line 20: suggest changing to ‘groups including A2ML1’

Made changes

Line 21: should it be ‘genomic’ instead of ‘genome’?

Made changes: genomic regions

Introduction:

Lines 27-31: not clear if these are all meant to be one sentence, these lines should be divided into separate sentences for clarity. The first sentence here is also somewhat confusing in relation to the use of terms like ‘pigs’ organisms’? do the author’s mean pigs’ physiological status?

Made changes

Line 50: remove full stop before reference

Made changes

Materials and Methods

Line 61: Why were Landrace and Duroc breeds used for this study, is capped hock more prevalent in these breeds as opposed to others?

We have selected these breeds in this study. In the future, we plan to study other.

Line 62: were all pigs used in this study of similar age, despite all being born in the same year. Was there any information about relatedness of the pigs, e.g. same boar?

All pigs were of the same year of birth, of the same age, and were kept in the same conditions.

Line 63: how were the pigs housed? Was there a difference in ground surface up to separation of pigs into groups that may have impacted capped hocks? Or was there a difference up to the end of the trial in the type of flooring the pigs were reared on?

There was no difference in the type of flooring. All pigs on the farm are raised under the same conditions.

Results

Line 102: gene names should be in italics

Manuscript Preparation was according to the Journal's Instructions for Authors (gene names should not be italicized)

Lines 103-108/ 116-120: what are the p-value significance of each of these pathways? And why is a pathway listed as ‘unclassified’ on line 107?

Unclassified are the functions which have not been studied yet. Agree, it is better to remove this from the text.

Added

Cadherin signaling pathway (p-value = 8.95e-02)

Wnt signaling pathway (p-value = 1.88e-01)

Notch signaling pathway (NUMB) (p-value = 2.55е-02)

Nicotinic acetylcholine receptor signaling pathway (p-value = 5.81е-02)

Line 118: remove ) from line

Made changes

Line 121: what is meant by ‘genome regions with strong outliers’? There are no p-values included within the results text or tables (including supplemental tables)??

To determine selection signatures we used smoothing FST statistics based on the proposed model with pure drift (Nicholson et al. 2002). Individual SNP FST values are grouped into genomic windows (contiguous regions) to determine smoothed FST values that identify genomic regions with high FST values. FST was performed by comparing pigs L_I vs L_II and D_I vs D_II. SNP regions with smoothed FST values above the 99th quantile were identified; the gene and QTL content of each region was analyzed in the Ensembl genome browser (Sscrofa 11.1) (https://www.ensembl.org/index.html). Accordingly, p-values are not further specified.

(Regions with strong outliers’ are regions with smoothed FST values above the 99th quantile)

Line 128: authors mention the importance of flooring type to capped hock but don’t mention flooring type used for the pigs in this study from birth within the methods

The pigs are kept on concrete slotted floors. This is the worldwide standard for breeding centers. The problem of capped hock is just related to the transition to the intensification of the breeding process of pigs and changes in conditions. But these are not the local conditions of one farm, but a complete restructuring of the pig breeding industry as a whole.

Line 168: should this be ‘capped hock’ and not ‘sarred hock’?

Made changes

Line 186: in which breed?

Made changes (in Landrace pigs)

Line 191/192: in which breed?

Made changes (in Duroc pigs)

Line 204: should this not read ‘highly productive pigs free from various defects’ as in they don’t have the defects?

Made changes (highly productive pigs free from various defects’)

Table 1 and supplemental tables: please define SSC abbreviation

Made changes (SSC – Sus scrofa chromosome)

Round 2

Reviewer 1 Report

I am still not satisfied with the changes in the manuscript, following are my major concerns:

(1). The author indicates that they did not perform case-control study, however, respectfully I disagree, their design, in my opinion, is still case-control (https://en.wikipedia.org/wiki/Case%E2%80%93control_study), therefore, underlying genetic structure in the population will affect their results and therefore, interpretation. 

(2). They mentioned in their response that, "То study population structure, we performed a singular value decomposition (SVD) decomposition of the GRM using the SVD function in R (Barker et al., 2001; VanRaden, 2008).", however, this information is not added in the materials and method section in the paper. Please also display and discuss the results of this step. 

(3). If they took into account "all external factors", please also mention it in the materials and method section as the statement concerning is still missing. 

Reviewer 2 Report

some of my original comments remain which should be amended within the manuscript text as follows:

  • Why were Landrace and Duroc breeds used for this study, the authors provide no rational for their use over other breeds within the manuscript text
  • if pigs were related or not (e.g. same boar used or a number of different boars) authors should include this information within the text 
  • the authors state the importance of ground surface to capped hock, yet don't state what ground surface (concrete slatted floors) was used for their study

Author Response

Reviewer 2.

All changes to manuscript text are highlighted in yellow

Why were Landrace and Duroc breeds used for this study, the authors provide no rational for their use over other breeds within the manuscript text

In many countries, including the Russian Federation, pig breeding is based on a three-level pyramidal structure. At the first level, Large White and Landrace breeds are used to obtain F1 hybrid sows. Next, F1 sows are mated to Duroc boars and the result is F2 final hybrids. Breeding work to improve Large White and Landrace is mainly aimed at reproductive traits, Duroc for growth and meat traits. For work, we chose Landrace and Duroс pigs as they differ in performance traits, but to some extent subject to capped hock.

Added in Materials and Methods

We chose Landrace and Duroс pigs as they differ in performance traits, but to some extent subject to capped hock.

if pigs were related or not (e.g. same boar used or a number of different boars) authors should include this information within the text

Genotype quality control and data filtering were performed using PLINK 1.9:

rel-cutoff 0.75 (excludes relatives) – added in Materials and Methods

the authors state the importance of ground surface to capped hock, yet don't state what ground surface (concrete slatted floors) was used for their study

Added in Materials and Methods

All pigs on the farm were raised under the same conditions, on concrete slotted floors.